# The continuous Bernoulli: fixing a pervasive error in variational autoencoders

**Gabriel Loaiza-Ganem**
Department of Statistics
Columbia University
gl2480@columbia.edu

**John P. Cunningham**
Department of Statistics
Columbia University
jpc2181@columbia.edu

## Abstract

Variational autoencoders (VAE) have quickly become a central tool in machine learning, applicable to a broad range of data types and latent variable models. By far the most common first step, taken by seminal papers and by core software libraries alike, is to model MNIST data using a deep network parameterizing a Bernoulli likelihood. This practice contains what appears to be and what is often set aside as a minor inconvenience: the pixel data is $[0, 1]$ valued, not $\{0, 1\}$ as supported by the Bernoulli likelihood. Here we show that, far from being a triviality or nuisance that is convenient to ignore, this error has profound importance to VAE, both qualitative and quantitative. We introduce and fully characterize a new $[0, 1]$-supported, single parameter distribution: the *continuous Bernoulli*, which patches this pervasive bug in VAE. This distribution is not nitpicking; it produces meaningful performance improvements across a range of metrics and datasets, including sharper image samples, and suggests a broader class of performant VAE.[1]

## 1 Introduction

Variational autoencoders (VAE) have become a central tool for probabilistic modeling of complex, high dimensional data, and have been applied across image generation [10], text generation [14], neuroscience [8], chemistry [9], and more. One critical choice in the design of any VAE is the choice of likelihood (decoder) distribution, which stipulates the stochastic relationship between latents and observables. Consider then using a VAE to model the MNIST dataset, by far the most common first step for introducing and implementing VAE. An apparently innocuous practice is to use a Bernoulli likelihood to model this $[0, 1]$-valued data (grayscale pixel values), in disagreement with the $\{0, 1\}$ support of the Bernoulli distribution. Though doing so will not throw an obvious type error, the implied object is no longer a coherent probabilistic model, due to a neglected normalizing constant. This practice is extremely pervasive in the VAE literature, including the seminal work of Kingma and Welling [20] (who, while aware of it, set it aside as an inconvenience), highly-cited follow up work (for example [25, 37, 17, 6] to name but a few), VAE tutorials [7, 1], including those in hugely popular deep learning frameworks such as PyTorch [32] and Keras [3], and more.

Here we introduce and fully characterize the *continuous Bernoulli* distribution (§3), both as a means to study the impact of this widespread modeling error, and to provide a proper VAE for $[0, 1]$-valued data. Before these details, let us ask the central question: *who cares*?

First, theoretically, ignoring normalizing constants is unthinkable throughout most of probabilistic machine learning: these objects serve a central role in restricted Boltzmann machines [36, 13], graphical models [23, 33, 31, 38], maximum entropy modeling [16, 29, 26], the "Occam's razor" nature of Bayesian models [27], and much more.

Second, one might suppose this error can be interpreted or fixed via data augmentation, binarizing data (which is also a common practice), stipulating a different lower bound, or as a nonprobabilistic model with a "negative binary cross-entropy" objective. §4 explores these possibilities and finds them wanting. Also, one might be tempted to call the Bernoulli VAE a toy model or a minor point. Let us avoid that trap: MNIST is likely the single most widely used dataset in machine learning, and VAE is quickly becoming one of our most popular probabilistic models.

Third, and most importantly, empiricism; §5 shows three key results: *(i)* as a result of this error, we show that the Bernoulli VAE significantly underperforms the continuous Bernoulli VAE across a range of evaluation metrics, models, and datasets; *(ii)* a further unexpected finding is that this performance loss is significant even when the data is close to binary, a result that becomes clear by consideration of continuous Bernoulli limits; and *(iii)* we further compare the continuous Bernoulli to beta likelihood and Gaussian likelihood VAE, again finding the continuous Bernoulli performant.

All together this work suggests that careful treatment of data type – neither ignoring normalizing constants nor defaulting immediately to a Gaussian likelihood – can produce optimal results when modeling some of the most core datasets in machine learning.

## 2 Variational autoencoders

Autoencoding variational Bayes [20] is a technique to perform inference in the model:

$$Z_n \sim p_0(z) \quad \text{and} \quad X_n|Z_n \sim p_\theta(x|z_n) \ , \ \text{for } n = 1, \ldots, N, \tag{1}$$

where each $Z_n \in \mathbb{R}^M$ is a local hidden variable, and $\theta$ are parameters for the likelihood of observables $X_n$. The prior $p_0(z)$ is conventionally a Gaussian $\mathcal{N}(0, I_M)$. When the data is binary, i.e. $x_n \in \{0, 1\}^D$, the conditional likelihood $p_\theta(x_n|z_n)$ is chosen to be $\mathcal{B}(\lambda_\theta(z_n))$, where $\lambda_\theta : \mathbb{R}^M \to \mathbb{R}^D$ is a neural network with parameters $\theta$. $\mathcal{B}(\lambda)$ denotes the product of $D$ independent Bernoulli distributions, with parameters $\lambda \in [0, 1]^D$ (in the standard way we overload $\mathcal{B}(\cdot)$ to be the univariate Bernoulli or the product of independent Bernoullis). Since direct maximum likelihood estimation of $\theta$ is intractable, variational autoencoders use VBEM [18], first positing a now-standard variational posterior family:

$$q_\phi\big((z_1, ..., z_n)|(x_1, ..., x_n)\big) = \prod_{n=1}^N q_\phi(z_n|x_n), \text{with } q_\phi(z_n|x_n) = \mathcal{N}\Big(m_\phi(x_n), \text{diag}\big(s_\phi^2(x_n)\big)\Big) \tag{2}$$

where $m_\phi : \mathbb{R}^D \to \mathbb{R}^M$, $s_\phi : \mathbb{R}^D \to \mathbb{R}_+^M$ are neural networks parameterized by $\phi$. Then, using stochastic gradient descent and the reparameterization trick, the evidence lower bound (ELBO) $\mathcal{E}(\theta, \phi)$ is maximized over both generative and posterior (decoder and encoder) parameters $(\theta, \phi)$:

$$\mathcal{E}(\theta, \phi) = \sum_{n=1}^N E_{q_\phi(z_n|x_n)}[\log p_\theta(x_n|z_n)] - KL(q_\phi(z_n|x_n)||p_0(z_n)) \le \log p_\theta\big((x_1, \ldots, x_N)\big). \tag{3}$$

### 2.1 The pervasive error in Bernoulli VAE

In the Bernoulli case, the reconstruction term in the ELBO is:

$$E_{q_\phi(z_n|x_n)}[\log p_\theta(x_n|z_n)] = E_{q_\phi(z_n|x_n)}\Big[\sum_{d=1}^D x_{n,d} \log \lambda_{\theta,d}(z_n) + (1-x_{n,d}) \log(1-\lambda_{\theta,d}(z_n))\Big] \tag{4}$$

where $x_{n,d}$ and $\lambda_{\theta,d}(z_n)$ are the $d$-th coordinates of $x_n$ and $\lambda_\theta(z_n)$, respectively. However, critically, Bernoulli likelihoods and the reconstruction term of equation 4 are commonly used for $[0, 1]$-valued data, which loses the interpretation of probabilistic inference. To clarify, hereafter we denote the Bernoulli distribution as $\tilde{p}(x|\lambda) = \lambda^x(1 - \lambda)^{1-x}$ to emphasize the fact that it is an unnormalized distribution (when evaluated over $[0, 1]$). We will also make this explicit in the ELBO, writing $\mathcal{E}(\tilde{p}, \theta, \phi)$ to denote that the reconstruction term of equation 4 is being used. When analyzing $[0, 1]$-valued data such as MNIST, the Bernoulli VAE has optimal parameter values $\theta^*(\tilde{p})$ and $\phi^*(\tilde{p})$; that is:

$$(\theta^*(\tilde{p}), \phi^*(\tilde{p})) = \underset{(\theta,\phi)}{\text{argmax}}\, \mathcal{E}(\tilde{p}, \theta, \phi). \tag{5}$$

# 3 $\mathcal{CB}$: the continuous Bernoulli distribution

In order to analyze the implications of this modeling error, we introduce the continuous Bernoulli, a novel distribution on $[0, 1]$, which is parameterized by $\lambda \in (0, 1)$ and defined by:

$$X \sim \mathcal{CB}(\lambda) \iff p(x|\lambda) \propto \tilde{p}(x|\lambda) = \lambda^x (1-\lambda)^{1-x}. \tag{6}$$

We fully characterize this distribution, deferring proofs and secondary properties to appendices.

**Proposition 1** ($\mathcal{CB}$ **density and mean**): The continuous Bernoulli distribution is well defined, that is, $0 < \int_0^1 \tilde{p}(x|\lambda)dx < \infty$ for every $\lambda \in (0, 1)$. Furthermore, if $X \sim \mathcal{CB}(\lambda)$, then the density function of $X$ and its expected value are:

$$p(x|\lambda) = C(\lambda)\lambda^x(1-\lambda)^{1-x}, \text{where } C(\lambda) = \begin{cases} \dfrac{2\tanh^{-1}(1-2\lambda)}{1-2\lambda} & \text{if } \lambda \neq 0.5 \\ 2 & \text{otherwise} \end{cases} \tag{7}$$

$$\mu(\lambda) := E[X] = \begin{cases} \dfrac{\lambda}{2\lambda-1} + \dfrac{1}{2\tanh^{-1}(1-2\lambda)} & \text{if } \lambda \neq 0.5 \\ 0.5 & \text{otherwise} \end{cases} \tag{8}$$

Figure 1 shows $\log C(\lambda)$, $p(x|\lambda)$, and $\mu(\lambda)$. Some notes warrant mention: *(i)* unlike the Bernoulli, the mean of the continuous Bernoulli is not $\lambda$; *(ii)* however, like for the Bernoulli, $\mu(\lambda)$ is increasing on $\lambda$ and goes to 0 or 1 when $\lambda$ goes to 0 or 1; *(iii)* the continuous Bernoulli is *not* a beta distribution (the main difference between these two distributions is how they concentrate mass around the extrema, see appendix 1 for details), nor any other $[0, 1]$-supported distribution we are aware of (including continuous relaxations such as the Gumbel-Softmax [28, 15], see appendix 1 for details); *(iv)* $C(\lambda)$ and $\mu(\lambda)$ are continuous functions of $\lambda$; and *(v)* the log normalizing constant $\log C(\lambda)$ is well characterized but numerically unstable close to $\lambda = 0.5$, so our implementation uses a Taylor approximation near that critical point to calculate $\log C(\lambda)$ to high numerical precision. **Proof**: See appendix 3.

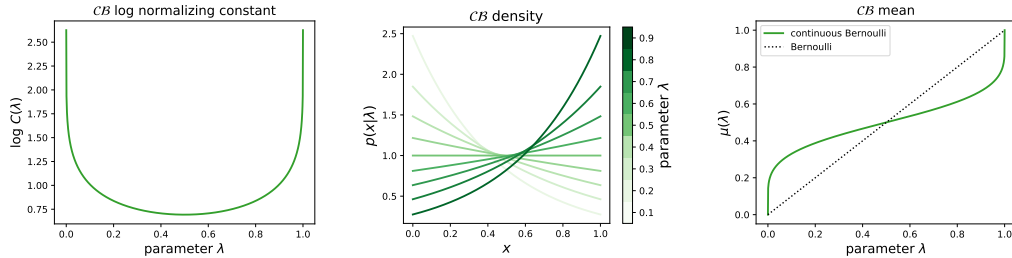

Figure 1: continuous Bernoulli log normalizing constant (left), pdf (middle) and mean (right).

**Proposition 2** ($\mathcal{CB}$ **additional properties**): The continuous Bernoulli forms an exponential family, has closed form variance, CDF, inverse CDF (which importantly enables easy sampling and the use of the reparameterization trick), characteristic function (and thus moment generating function too) and entropy. Also, the KL between two continuous Bernoulli distributions also has closed form and $C(\lambda)$ is convex. Finally, the continuous Bernoulli admits a conjugate prior which we call the C-Beta distribution. See appendix 2 for details. **Proof**: See appendix 3.

**Proposition 3** ($\mathcal{CB}$ **Bernoulli limit**): $\mathcal{CB}(\lambda) \xrightarrow{\lambda \to 0} \delta(0)$ and $\mathcal{CB}(\lambda) \xrightarrow{\lambda \to 1} \delta(1)$ in distribution; that is, the continuous Bernoulli becomes a Bernoulli in the limit. **Proof**: See appendix 3.

This proposition might at a first glance suggest that, as long as the estimated parameters are close to 0 or 1 (which should happen when the data is close to binary), then the practice of erroneously applying a Bernoulli VAE should be of little consequence. However, the above reasoning is wrong, as it ignores the shape of $\log C(\lambda)$: as $\lambda \to 0$ or $\lambda \to 1$, $\log C(\lambda) \to \infty$ (Figure 1, left). Thus, if data is close to binary, the term neglected by the Bernoulli VAE becomes even more important, the exact opposite conclusion than the naive one presented above.

**Proposition 4** ($\mathcal{CB}$ **normalizing constant bound**): $C(\lambda) \geq 2$, with equality if and only if $\lambda = 0.5$. And thus it follows that, for any $x, \lambda$, we have $\log p(x|\lambda) > \log \tilde{p}(x|\lambda)$. **Proof**: See appendix 3.

This proposition allows us to interpret $\mathcal{E}(\tilde{p}, \theta, \phi)$ as a *lower* lower bound of the log likelihood (§4).

**Proposition 5** ($\mathcal{CB}$ **maximum likelihood**): For an observed sample $x_1, \ldots, x_N \sim_{iid} \mathcal{CB}(\lambda)$, the maximum likelihood estimator $\hat{\lambda}$ of $\lambda$ is such that $\mu(\hat{\lambda}) = \frac{1}{N} \sum_n x_n$. **Proof**: See appendix 3.

Beyond characterizing a novel and interesting distribution, these propositions now allow full analysis of the error in applying a Bernoulli VAE to the wrong data type.

## 4 The continuous Bernoulli VAE, and why the normalizing constant matters

We define the continuous Bernoulli VAE analogously to the Bernoulli VAE:

$$Z_n \sim \mathcal{N}(0, I_M) \quad \text{and} \quad X_n|Z_n \sim \mathcal{CB}\left(\lambda_\theta(z_n)\right) \ , \ \text{for } n = 1, \ldots, N \tag{9}$$

where again $\lambda_\theta : \mathbb{R}^M \to \mathbb{R}^D$ is a neural network with parameters $\theta$, and $\mathcal{CB}(\lambda)$ now denotes the product of $D$ independent *continuous* Bernoulli distributions. Operationally, this modification results only in a change to the optimized objective; for clarity we compare the ELBO of the continuous Bernoulli VAE (top), $\mathcal{E}(p, \theta, \phi)$, to the Bernoulli VAE (bottom):

$$\mathcal{E}(p, \theta, \phi) = \sum_{n=1}^{N} -KL(q_\phi || p_0) + E_{q_\phi}\left[\sum_{d=1}^{D} x_{n,d} \log \lambda_{\theta,d}(z_n) + (1 - x_{n,d}) \log(1 - \lambda_{\theta,d}(z_n)) + \log C(\lambda_{\theta,d}(z_n))\right]$$

$$\mathcal{E}(\tilde{p}, \theta, \phi) = \sum_{n=1}^{N} -KL(q_\phi || p_0) + E_{q_\phi}\left[\sum_{d=1}^{D} x_{n,d} \log \lambda_{\theta,d}(z_n) + (1 - x_{n,d}) \log(1 - \lambda_{\theta,d}(z_n))\right],$$

Analogously, we denote $\theta^*(p)$ and $\phi^*(p)$ as the maximizers of the continuous Bernoulli ELBO:

$$(\theta^*(p), \phi^*(p)) = \underset{(\theta, \phi)}{\operatorname{argmax}} \mathcal{E}(p, \theta, \phi). \tag{10}$$

Immediately, a number of potential interpretations for the Bernoulli VAE come to mind, some of which have appeared in literature. We analyze each in turn.

### 4.1 Changing the data, model or training objective

One obvious workaround is to simply binarize any $[0, 1]$-valued data (MNIST pixel values or otherwise), so that it accords with the Bernoulli likelihood [24], a practice that is commonly followed (e.g. [34, 2, 28, 15, 12]). First, modifying data to fit a model, particularly an unsupervised model, is fundamentally problematic. Second, many $[0, 1]$-valued datasets are heavily degraded by binarization (see appendices for CIFAR-10 samples), indicating major practical limitations. Another workaround is to change the likelihood of the model to a proper $[0, 1]$-supported distribution, such as a beta or a truncated Gaussian. In §5 we include comparisons against a VAE with a beta distribution likelihood (we also made comparisons against a truncated Gaussian but found this to severely underperform all the alternatives). Gulrajani et al. [11] use a discrete distribution over the $256$ possible pixel values. Knoblauch et al. [22] study changing the reconstruction and/or the KL term in the ELBO. While their main focus is to obtain more robust inference, they provide a framework in which the Bernoulli VAE corresponds simply to a different (nonprobabilistic) loss. In this perspective, empirical results must determine the adequacy of this objective; §5 shows the Bernoulli VAE to underperform its proper probabilistic counterpart across a range of settings. Finally, note that none of these alternatives provide a way to understand the effect of using Bernoulli likelihoods on $[0, 1]$-valued data.

### 4.2 Data augmentation

Since the expectation of a Bernoulli random variable is precisely its parameter, the Bernoulli VAE might (erroneously) be assumed to be equivalent to a continuous Bernoulli VAE on an infinitely augmented dataset, obtained by sampling binary data whose mean is given by the observed data; indeed this idea is suggested by Kingma and Welling [20][2]. This interpretation does not hold[3]; it would result in a reconstruction term as in the first line in the equation below, while a correct Bernoulli

VAE on the augmented dataset would have a reconstruction term given by the second line (not equal, as the order of expectation can not be switched since $q_\phi$ depends on $X_r$ on the second line):

$$E_{z_n \sim q_\phi(z_n|x_n)} \Big[ E_{X_n \sim \mathcal{B}(x_n)} \Big[ \sum_{d=1}^{D} X_{n,d} \log \lambda_{\theta,d}(z_n) + (1 - X_{n,d}) \log \lambda_{\theta,d}(z_n) \Big] \Big] \tag{11}$$

$$\neq E_{X_n \sim \mathcal{B}(x_n)} \Big[ E_{z_n \sim q_\phi(z_n|X_n)} \Big[ \sum_{d=1}^{D} X_{n,d} \log \lambda_{\theta,d}(z_n) + (1 - X_{n,d}) \log \lambda_{\theta,d}(z_n) \Big] \Big].$$

### 4.3 Bernoulli VAE as a lower lower bound

Because the continuous Bernoulli ELBO and the Bernoulli ELBO are related by:

$$\mathcal{E}(\tilde{p}, \theta, \phi) + \sum_{n=1}^{N} \sum_{d=1}^{D} E_{z_n \sim q_\phi(z_n|x_n)}[\log C(\lambda_{\theta,d}(z_n))] = \mathcal{E}(p, \theta, \phi) \tag{12}$$

and recalling Proposition 4, since $\log 2 > 0$, we get that $\mathcal{E}(\tilde{p}, \theta, \phi) < \mathcal{E}(p, \theta, \phi)$. That is, using the Bernoulli VAE results in optimizing an even-lower bound of the log likelihood than using the continuous Bernoulli ELBO. Note however that unlike the ELBO, $\mathcal{E}(\tilde{p}, \theta, \phi)$ is not tight even if the approximate posterior matches the true posterior.

### 4.4 Mean parameterization

The conventional maximum likelihood estimator for a Bernoulli, namely $\hat{\lambda}_{\mathcal{B}} = \frac{1}{N} \sum_n x_n$, maximizes $\tilde{p}(x_1, ..., x_N | \lambda)$ regardless of whether data is $\{0, 1\}$ and $[0, 1]$. As a thought experiment, consider $x_1, \ldots, x_N \sim_{iid} \mathcal{CB}(\lambda)$. Proposition 5 tells us that the correct maximum likelihood estimator, $\hat{\lambda}_{\mathcal{CB}}$ is such that $\mu(\hat{\lambda}_{\mathcal{CB}}) = \frac{1}{N} \sum_n x_n$, where $\mu$ is the $\mathcal{CB}$ mean of equation 8. Thus, while using $\hat{\lambda}_{\mathcal{B}}$ is incorrect, one can (surprisingly) still recover the correct maximum likelihood estimator via $\hat{\lambda}_{\mathcal{CB}} = \mu^{-1}(\hat{\lambda}_{\mathcal{B}})$. One might then (wrongly) think that training a Bernoulli VAE, and then subsequently transforming the decoder parameters with $\mu^{-1}$, would be equivalent to training a continuous Bernoulli VAE; that is, $\lambda_{\theta^*(p)}$ might be equal to $\mu^{-1}(\lambda_{\theta^*(\tilde{p})})$. This reasoning is incorrect: the KL term in the ELBO implies that $\lambda_{\theta^*(p)}(z_n) \neq \mu^{-1}(x_n)$, and so too $\lambda_{\theta^*(\tilde{p})}(z_n) \neq x_n$, and as such $\lambda_{\theta^*(p)} \neq \mu^{-1}(\lambda_{\theta^*(\tilde{p})})$. In fact, our experiments will show that despite this flawed reasoning, applying this transformation can recover some, but not all, of the performance loss from using a Bernoulli VAE.

## 5 Experiments

We have introduced the continuous Bernoulli distribution to give a proper probabilistic VAE for $[0, 1]$-valued data. The essential question that we now address is how much, if any, improvement we achieve by making this choice.

### 5.1 MNIST

One frequently noted shortcoming of VAE (and Bernoulli VAE on MNIST in particular) is that samples from this model are blurry. As noted, the convexity of $\log C(\lambda)$ can be seen as regularizing sample values from the VAE to be more extremal; that is, sharper. As such we first compared samples from a trained continuous Bernoulli VAE against samples from the MNIST dataset itself, from a trained Bernoulli VAE, and from a trained Gaussian VAE, that is, the usual VAE model with a decoder likelihood $p_\theta(x|z) = \mathcal{N}(\eta_\theta(z), \sigma_\theta^2(z))$, where we use $\eta$ to avoid confusion with $\mu$ of equation 8. These samples are shown in Figure 2. In each case, as is standard, we show the parameter output by the generative/decoder network for a given latent draw: $\lambda_{\theta^*(p)}(z)$ for the $\mathcal{CB}$ VAE, $\lambda_{\theta^*(\tilde{p})}(z)$ for the $\mathcal{B}$ VAE, and $\eta_{\theta^*}(z)$ for the $\mathcal{N}$ VAE. Qualitatively, the continuous Bernoulli VAE achieves considerably superior samples vs the Bernoulli or Gaussian VAE, owing to the effect of $\log C(\lambda)$ pushing the decoder toward sharper images. Further samples are in the appendix. Dai and Wipf [4] consider why Gaussian VAE produce blurry images; we point out that our work (considering the likelihood) is orthogonal to theirs (considering the data manifold).

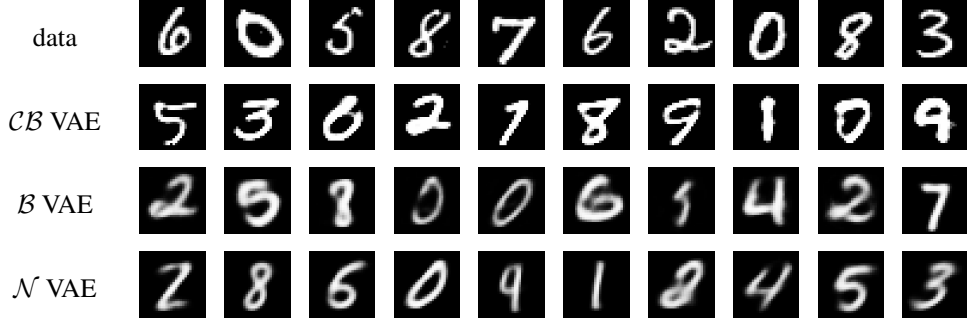

data

$\mathcal{CB}$ VAE

$\mathcal{B}$ VAE

$\mathcal{N}$ VAE

Figure 2: Samples from MNIST, continuous Bernoulli VAE, Bernoulli VAE, and Gaussian VAE.

## 5.2 Warped MNIST datasets

The most common justification for the Bernoulli VAE is that MNIST pixel values are 'close' to binary. An important study is thus to ask how the performance of continuous Bernoulli VAE vs the Bernoulli VAE changes as a function of this 'closeness.' We formalize this concept by introducing a warping function $f_\gamma(x)$ that, depending on the warping parameter $\gamma$, transforms individual pixel values to produce a dataset that is anywhere from fully binarized (every pixel becomes $\{0, 1\}$) to fully degraded (every pixel becomes $0.5$). Figure 3 shows $f_\gamma$ for different values of $\gamma$, and the (rather uninformative) warping equation appears next to Figure 3.

Importantly, $\gamma = 0$ corresponds to an unwarped dataset, i.e., the original MNIST dataset. Further, note that negative values of $\gamma$ warp pixel values towards being more binarized, completely binarizing it in the case where $\gamma = -0.5$, whereas positive values of $\gamma$ push the pixel values towards $0.5$, recovering constant data at $\gamma = 0.5$. We then train our competing VAE models on the datasets induced by different values of $\gamma$ and compare the difference in performance as $\gamma$ changes. Note importantly that, because $\gamma$ induces different datasets, performance values should primarily be compared between VAE models at each $\gamma$ value; the trend as a function of $\gamma$ is not of particular interest.

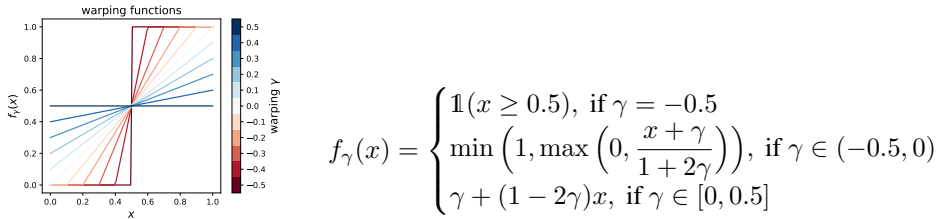

$$f_\gamma(x) = \begin{cases} \mathbb{1}(x \geq 0.5), & \text{if } \gamma = -0.5 \\ \min\left(1, \max\left(0, \dfrac{x + \gamma}{1 + 2\gamma}\right)\right), & \text{if } \gamma \in (-0.5, 0) \\ \gamma + (1 - 2\gamma)x, & \text{if } \gamma \in [0, 0.5] \end{cases}$$

Figure 3: $f_\gamma$ for different $\gamma$ values.

Figure 4 shows the results of various models applied to these different datasets (all values are an average of 10 separate training runs). The same neural network architectures are used across this figure, with architectural choices that are quite standard (detailed in appendix 4, along with training hyperparameters). The left panel shows ELBO values. In dark blue is the continuous Bernoulli VAE ELBO, namely $\mathcal{E}(p, \theta^*(p), \phi^*(p))$. In light blue is the same ELBO when evaluated on a trained Bernoulli VAE: $\mathcal{E}(p, \theta^*(\tilde{p}), \phi^*(\tilde{p}))$. Most importantly, note the $\gamma = 0$ values; the continuous Bernoulli VAE achieves a 300 nat improvement over the Bernoulli VAE. This finding supports the previous qualitative finding and the theoretical motivation for this work: significant quantitative gains are achieved via the continuous Bernoulli model. This finding remains true across a range of $\gamma$ (dark blue above light blue in Figure 4), indicating that regardless of how 'close' to binary a dataset is, the continuous Bernoulli is a superior VAE model.

One might then wonder if the continuous Bernoulli is outperforming simply because the Bernoulli needs a mean correction. We thus apply $\mu^{-1}$, namely the map from Bernoulli to continuous Bernoulli maximum likelihood estimators (equation 8 and §4.4), and evaluate the same ELBO on $\mu^{-1}(\lambda_{\theta^*(\tilde{p})})$ as the decoder shown in light red (Figure 4, left). This result, which is only achieved via the introduction of the continuous Bernoulli, shows two important findings: first, that indeed some improvement over

the Bernoulli VAE is achieved by post hoc correction to a continuous Bernoulli parameterization; but second, that this transformation is still inadequate to achieve the full performance of the continuous Bernoulli VAE.

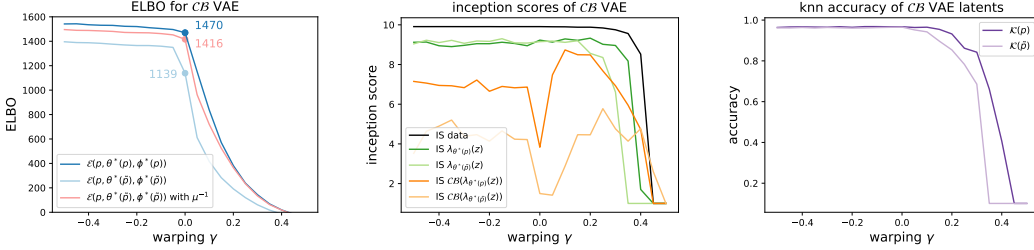

Figure 4: Continuous Bernoulli comparisons against Bernoulli VAE. See text for details.

We also note that we ran the same experiment with log likelihood instead of ELBO (using importance weighted estimates with $k = 100$ samples; see Burda et al. [2]), and the same results held (up to small numerical differences; these traces are omitted for figure clarity). We also ran the same experiment for the $\beta$-VAE [12], sweeping a range of $\beta$ values, and the same results held (see appendix 5.1). To make sure that the discrepancy between the continuous Bernoulli and Bernoulli is not due to the approximate posterior not being able to adequately approximate the true posterior, we ran the same experiments with flexible posteriors using normalizing flows [34] and found the discrepancy to become even larger (see appendix 5.1).

It is natural to then wonder if this performance is an artifact of ELBO and log likelihood; thus, we also evaluated the same datasets and models using different evaluation metrics. In the middle panel of Figure 4, we use the inception score [35] to measure sample quality produced by the different models (higher is better). Once again, we see that including the normalizing constant produces better samples (dark traces / continuous Bernoulli lie above light traces / Bernoulli). We include that comparison on both the decoder parameters $\lambda$ (dark and light green) and also samples from distributions indexed by those parameters (dark and light orange). One can imagine a variety of other parameter transformations that may be of interest; we include several in appendix 5.1, where again we find that the continuous Bernoulli VAE outperforms its Bernoulli counterpart.

In the right panel of Figure 4, to measure usefulness of the latent representations of these models, we compute $m_{\phi^*(p)}(x_n)$ and $m_{\phi^*(\tilde{p})}(x_n)$ (note that $m$ is the variational posterior mean from equation 2 and not the continuous Bernoulli mean) for training data and use a 15-nearest neighbor classifier to predict the test labels. The right panel of Figure 4 shows the accuracy of the classifiers (denoted $\mathcal{K}(\phi)$) as a function of $\gamma$. Once again, the continuous Bernoulli VAE outperforms the Bernoulli VAE.

Now that the continuous Bernoulli VAE gives us a proper model on $[0, 1]$, we can also propose other natural models. Here we introduce and compare against the beta distribution VAE (not $\beta$-VAE [12]); as the name implies, the generative likelihood is $\text{Beta}(\alpha_\theta(z), \beta_\theta(z))$. We repeated the same warped MNIST experiments using Gaussian VAE and beta distribution VAE, both including and ignoring the normalizing constants of those distributions, as an analogy to the continuous Bernoulli and Bernoulli distributions. First, Figure 5 shows again that that ignoring the normalizing constant hurts performance in every metric and model (dark above light). Second, interestingly, we find that the continuous Bernoulli VAE outperforms both the beta distribution VAE and the Gaussian VAE in terms of inception scores, and that the beta distribution VAE dominates in terms of ELBO. This figure clarifies that the continuous Bernoulli and beta distribution are to be preferred over the Gaussian for VAE applied to $[0, 1]$ valued data, and that ignoring normalizing constants is indeed unwise.

A few additional notes warrant mention on Figure 5. Unlike with the continuous Bernoulli, we should not expect the Gaussian and beta normalizing constants to go to infinity as extrema are reached, so we do not observe the same patterns with respect to $\gamma$ as we did with the continuous Bernoulli. Note also that the lower lower bound property of ignoring normalizing constants does not hold in general, as it is a direct consequence of the continuous Bernoulli log normalizing constant being nonnegative.

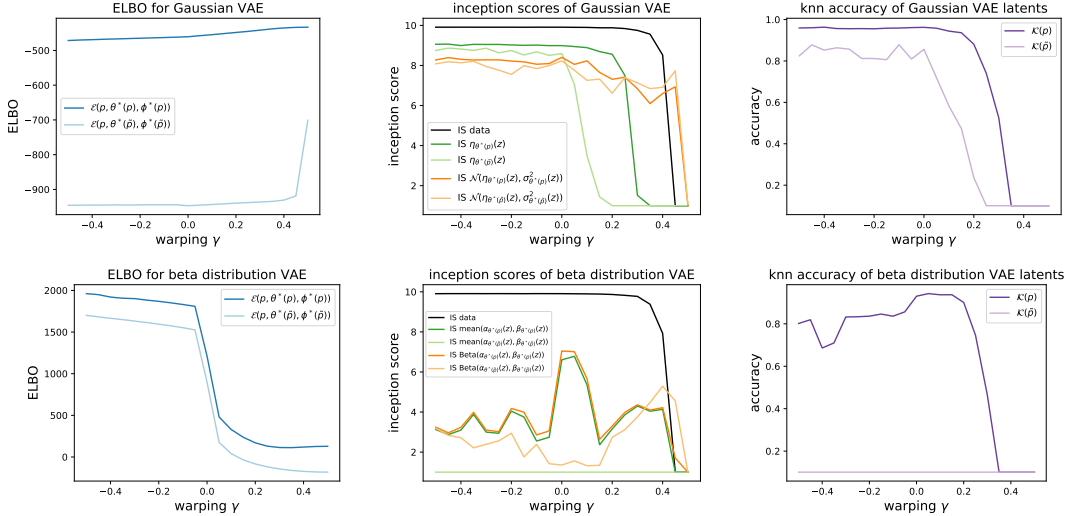

Figure 5: Gaussian (top panels) and beta (bottom panels) distributions comparisons between including and excluding the normalizing constants. Left panels show ELBOs, middle panels inceptions scores, and right panels 15-nearest neighbors accuracy.

Table 1: Comparisons of training with and without normalizing constants for CIFAR-10. For connection to the panels in Figures 4 and 5, column headers are colored accordingly.

| distribution | objective | map | $\mathcal{E}(p, \theta^*, \phi^*)$ | IS w/ samples | IS w/ parameters | $\mathcal{K}(\phi^*)$ |
|---|---|---|---|---|---|---|
| $\mathcal{CB}/\mathcal{B}$ | $\mathcal{E}(p, \theta, \phi)$ | · | **1007** | 1.15 | 2.31 | **0.43** |
| | $\mathcal{E}(\tilde{p}, \theta, \phi)$ | $\mu^{-1}$ | 916 | **1.49** | **4.55** | 0.42 |
| | $\mathcal{E}(\tilde{p}, \theta, \phi)$ | · | 475 | 1.41 | 1.39 | 0.42 |
| Gaussian | $\mathcal{E}(p, \theta, \phi)$ | · | **1891** | **1.86** | **3.04** | **0.42** |
| | $\mathcal{E}(\tilde{p}, \theta, \phi)$ | · | -42411 | 1.24 | 1.00 | 0.1 |
| beta | $\mathcal{E}(p, \theta, \phi)$ | · | **3121** | **2.98** | **4.07** | **0.47** |
| | $\mathcal{E}(\tilde{p}, \theta, \phi)$ | · | -38913 | 1.39 | 1.00 | 0.1 |

## 5.3 CIFAR-10

We repeat the same experiments as in the previous section on the CIFAR-10 dataset (without common preprocessing that takes the data outside $[0, 1]$ support), a dataset often considered to be a bad fit for Bernoulli VAE. For brevity we evaluated only the non-warped data $\gamma = 0$, leading to the results shown in Table 1 (note the colored column headers, to connect to the panels in Figures 4,5). The top section shows results for the continuous Bernoulli VAE (first row, top), the Bernoulli VAE (third row, top), and the Bernoulli VAE with a continuous Bernoulli inverse map $\mu^{-1}$ (second row, top). Across all metrics – ELBO, inception score with parameters $\lambda$, inception score with continuous Bernoulli samples given $\lambda$, and $k$ nearest neighbors – the Bernoulli VAE is suboptimal. Interestingly, unlike in MNIST, here we see that using the continuous Bernoulli parameter correction $\mu^{-1}$ (§4.4) to a Bernoulli VAE is optimal under some metrics. Again we note that this is a result belonging to the continuous Bernoulli (the parameter correction is derived from equation 8), so even these results emphasize the importance of the continuous Bernoulli.

We then repeat the same set of experiments for Gaussian and beta distribution VAE (middle and bottom sections of Table 1). Again, ignoring normalizing constants produces significant performance loss across all metrics. Comparing metrics across the continuous Bernoulli, Gaussian, and beta sections, we see again that the Gaussian VAE is suboptimal across all metrics, with the optimal distribution being the continuous Bernoulli or beta distribution VAE, depending on the metric.

## 5.4 Parameter estimation with EM

Finally, one might wonder if the performance improvements of the continuous Bernoulli VAE over the Bernoulli VAE and its corrected version with $\mu^{-1}$ are merely an artifact of not having access to the log likelihood and having to optimize the ELBO instead. In this section we show, empirically, that the answer is no. We consider estimating the parameters of a mixture of continuous Bernoulli distributions, of which the VAE can be thought of as a generalization with infinitely many components. We proceed as follows: We randomly set a mixture of continuous Bernoulli distributions, $p_{true}$, with $K$ components in 50 dimensions (independent of each other) and sample from this mixture 10000 times in order to generate a simulated dataset. We then use the EM algorithm [5] to estimate the mixture coefficients and the corresponding continuous Bernoulli parameters, first using a continuous Bernoulli likelihood (correct), and second using a Bernoulli likelihood (incorrect). We then measure how close the estimated parameters are from ground truth. To avoid a (hard) optimization problem over permutations, we measure closeness with KL divergence between the true distribution $p_{true}$ and the estimated $p_{est}$.

The results of performing the procedure described above 10 times and averaging the KL values (over these 10 repetitions), along with standard errors, are shown in Figure 6. First, we can see that when using the correct continuous Bernoulli likelihood, the EM algorithm correctly recovers the true distribution. We can also see that, as the number of mixture components $K$ gets larger, ignoring the normalizing constant results in worse performance, even after correcting with $\mu^{-1}$, which helps but does not fully remedy the situation (except at $K = 1$, as noted in §4.4).

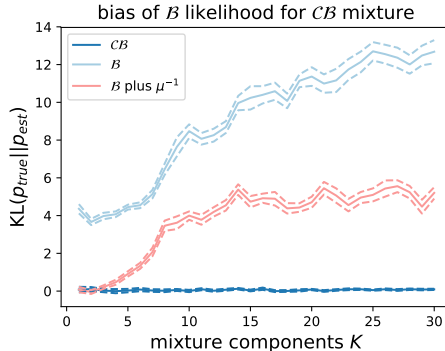

Figure 6: Bias of the EM algorithm to estimate $\mathcal{CB}$ parameters when using a $\mathcal{CB}$ likelihood (dark blue), $\mathcal{B}$ likelihood (light blue) and a $\mathcal{B}$ likelihood plus a $\mu^{-1}$ correction (light red).

## 6 Conclusions

In this paper we introduce and characterize a novel probability distribution – the continuous Bernoulli – to study the effect of using a Bernoulli VAE on $[0, 1]$-valued intensity data, a pervasive error in highly cited papers, publicly available implementations, and core software tutorials alike. We show that this practice is equivalent to ignoring the normalizing constant of a continuous Bernoulli, and that doing so results in significant performance decrease in the qualitative appearance of samples from these models, the ELBO (approximately 300 nats), the inception score, and in terms of the latent representation (via $k$ nearest neighbors). Several surprising findings are shown, including: *(i)* that some plausible interpretations of ignoring a normalizing constant are in fact wrong; *(ii)* the (possibly counterintuitive) fact that this normalizing constant is most critical when data is near binary; and *(iii)* that the Gaussian VAE often underperforms VAE models with the appropriate data type (continuous Bernoulli or beta distributions).

Taken together, these findings suggest an important potential role for the continuous Bernoulli distribution going forward. On this point, we note that our characterization of the continuous Bernoulli properties (such as its ease of reparameterization, likelihood evaluation, and sampling) make it compatible with the vast array of VAE improvements that have been proposed in the literature, including flexible posterior approximations [34, 21], disentangling [12], discrete codes [28, 15], variance control strategies [30], and more.

**Acknowledgments**

We thank Yixin Wang, Aaron Schein, Andy Miller, and Keyon Vafa for helpful conversations, and the Simons Foundation, Sloan Foundation, McKnight Endowment Fund, NIH NINDS 5R01NS100066, NSF 1707398, and the Gatsby Charitable Foundation for support.

## Footnotes

[1]Our code is available at https://github.com/cunningham-lab/cb.

[2]see the comments in `https://openreview.net/forum?id=33X9fd2-9FyZd`

[3]see `http://ruishu.io/2018/03/19/bernoulli-vae/` for a looser lower bound interpretation

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
