[Supplementary Material]

# Appendix for "The continuous Bernoulli: fixing a pervasive error in variational autoencoders"

## Appendix 1: differences between the continuous Bernoullii, the beta distribution, and continuous relaxations

First we discuss the beta distribution. While the beta, having two parameters, is more flexible than the continuous Bernoulli, these distributions have a key difference: concentration of mass at the extrema. For a given continuous Bernoulli parameter $\lambda$, beta distribution parameters $\alpha_\lambda$ and $\beta_\lambda$ can be found so that the corresponding distributions are very close (by matching the mean and variance), but this closeness does not happen at the extrema. To see this, we denote a beta density as $p(x|\alpha, \beta)$. We have:

$$\lim_{x \to 0} \log \frac{p(x|\lambda)}{p(x|\alpha, \beta)} = \lim_{x \to 0} \log C(\lambda) + \log B(\alpha, \beta) + x \log \lambda + (1-x) \log(1-\lambda)$$
$$- (\alpha - 1) \log x - (\beta - 1) \log(1-x)$$
$$= \begin{cases} -\infty & \text{if } \alpha < 1 \\ \log C(\lambda) + \log(1-\lambda) + \log B(\alpha, \beta) & \text{if } \alpha = 1 \\ \infty & \text{if } \alpha > 1 \end{cases}$$

where $B(\cdot, \cdot)$ is the beta function. This implies that:

$$\lim_{x \to 0} \frac{p(x|\lambda)}{p(x|\alpha, \beta)} = \begin{cases} 0 & \text{if } \alpha < 1 \\ C(\lambda)(1-\lambda)B(\alpha, \beta) & \text{if } \alpha = 1 \\ \infty & \text{if } \alpha > 1 \end{cases}$$

So that the continuous Bernoulli and the beta can place a comparable amount of mass around 0 only for $\alpha = 1$, otherwise the continuous Bernoulli places more mass if $\alpha > 1$ and less if $\alpha < 1$. In particular, for $\lambda < 0.5$ we have that $\alpha_\lambda < 1$, so that the beta distribution places much more mass around 0 than the continuous Bernoulli to which it is very similar, as can be seen in Figure 1. Note that this does not imply that the continuous Bernoulli is not placing most of its mass around 0, just not as much as the beta. This key insight highlights that even "similar looking" continuous Bernoullis and betas behave considerably differently at the extrema, which is precisely the most important part of the densities when modeling almost binary data (such as MNIST, which while almost binary, does have grayscale pixels). Empirically, we find that the beta distributed VAE produces means that are less extremal (i.e. grayer images, as seen in Figure 8), which implies that each beta was shifted away from 0 and 1 to reduce adding too much mass to the extrema (precisely the effect of Figure 1). Note than an analogous discussion holds around 1, since:

$$\lim_{x \to 1} \frac{p(x|\lambda)}{p(x|\alpha, \beta)} = \begin{cases} 0 & \text{if } \beta < 1 \\ C(\lambda)\lambda B(\alpha, \beta) & \text{if } \beta = 1 \\ \infty & \text{if } \beta > 1 \end{cases}$$

Finally, we also point out that the continuous Bernoulli is *not* a continuous relaxation of a Bernoulli. Continuous relaxations, such as the Gumbel-Softmax [28, 15], can approximate (i.e. converge in distribution to) any discrete distribution, usually by annealing a temperature hyperparameter. In our

Figure 1: Behavior of continuous Bernoulli against similar beta around 0.

setting, this property means that a continuous relaxation can approximate $\mathcal{B}(\lambda)$ for any $\lambda \in [0, 1]$, while the continuous Bernoulli can only do so for $\lambda \in \{0, 1\}$. Furthermore, unlike the continuous Bernoulli, the beta distribution and continuous relaxations cannot be used as a device of study for the effect of using Bernoulli likelihoods with $[0, 1]$-valued data.

## Appendix 2: additional properties of the continuous Bernoulli distribution

In this section we list some properties of the continuous Bernoulli distribution that are not directly useful for our results but that are nonetheless important to characterize this distribution.

**Proposition 2 ($\mathcal{CB}$ additional properties)**: The continuous Bernoulli forms an exponential family with sufficient statistic $T(x) = x$ and natural parameter $\log(\lambda/(1-\lambda))$. Furthermore, if $X \sim \mathcal{CB}(\lambda)$, the following hold:

1. The variance of $X$ is:
$$\text{var}(X) = \begin{cases} \dfrac{(\lambda - 1)\lambda}{(1 - 2\lambda)^2} + \dfrac{1}{(2\tanh^{-1}(1 - 2\lambda))^2} & \text{if } \lambda \neq 0.5 \\ 1/12 & \text{otherwise} \end{cases}$$

2. The cumulative distribution function of $X$ is:
$$F(x|\lambda) = \begin{cases} \dfrac{\lambda^x (1 - \lambda)^{1-x} + \lambda - 1}{2\lambda - 1} & \text{if } \lambda \neq 0.5 \\ x & \text{otherwise} \end{cases}$$

3. The inverse cumulative distribution function of $X$ is:
$$F^{-1}(u|\lambda) = \begin{cases} \dfrac{\log(u(2\lambda - 1) + 1 - \lambda) - \log(1 - \lambda)}{\log(\lambda) - \log(1 - \lambda)} & \text{if } \lambda \neq 0.5 \\ u & \text{otherwise} \end{cases}$$

   and thus we immediately have the reparameterization $X = F^{-1}(U)$ (in distribution), where $U \sim \text{Uniform}(0, 1)$.

4. The characteristic function of $X$ is:
$$\varphi_X(t) = \begin{cases} C(\lambda) \dfrac{i(1 - \lambda e^{it} - \lambda)}{t + 2it\tanh^{-1}(1 - 2\lambda)} & \text{if } \lambda \neq 0.5 \text{ and } t \neq 0 \\ \dfrac{e^{it} - 1}{it} & \text{if } \lambda = 0.5 \text{ and } t \neq 0 \\ 1 & \text{if } t = 0 \end{cases}$$

5. The entropy of $X$ is:
$$H(X) = \mu(\lambda) \log \frac{1 - \lambda}{\lambda} - \log C(\lambda) - \log(1 - \lambda)$$

6. The KL divergence between two continuous Bernoulli distributions is:
$$KL\big(p(x|\lambda_1)||p(x|\lambda_2)\big) = \mu(\lambda_1)\log\frac{\lambda_1(1-\lambda_2)}{\lambda_2(1-\lambda_1)} + \log\frac{C(\lambda_1)(1-\lambda_1)}{C(\lambda_2)(1-\lambda_2)}$$

7. $C(\lambda)$ is convex.

8. The following $(0,1)$-supported distribution, which we call C-Beta and denote by $p(\lambda|\alpha,\beta,\nu)$, where $\alpha > 0$, $\beta > 0$ and $\nu \geq 0$, is a proper distribution and is a conjugate prior to the continuous Bernoulli:
$$p(\lambda|\alpha, \beta\,\nu) \propto \lambda^{\alpha-1}(1-\lambda)^{\beta-1}C(\lambda)^{\nu}$$

We leave the study of further details of this distribution for future work.

## Appendix 3: proofs of propositions

In this section we prove all the properties about the continuous Bernoulli distribution. Note that if $X \sim \mathcal{CB}(0.5)$ then $X$ follows a uniform distribution on $[0,1]$, which immediately proves all the cases when $\lambda = 0.5$ in all the propositions. In what follows we assume that $\lambda \neq 0.5$.

**Proof of proposition 1 ($\mathcal{CB}$ density and mean)**:

We define the function $\tilde{F}(\cdot|\lambda) : [0,1] \to [0,1]$ as:
$$\tilde{F}(x|\lambda) = -\frac{\lambda^x(1-\lambda)^{1-x} + \lambda - 1}{2\tanh^{-1}(1-2\lambda)}$$

It is straightforward to verify that $\tilde{F}'(x|\lambda) = \tilde{p}(x|\lambda)$, so that for every $x \in [0,1]$:
$$\int_0^x \lambda^s(1-\lambda)^{1-s}ds = \tilde{F}(x|\lambda) - \tilde{F}(0|\lambda)$$

In particular:
$$0 < \frac{1}{C(\lambda)} = \int_0^1 \lambda^s(1-\lambda)^{1-s}ds = \tilde{F}(1|\lambda) - \tilde{F}(0|\lambda) = \frac{1-2\lambda}{2\tanh^{-1}(1-2\lambda)} < \infty$$

And thus $p(x|\lambda) = C(\lambda)\tilde{p}(x|\lambda)$. Furthermore:
$$\mu(\lambda) = \int_0^1 C(\lambda)x\lambda^x(1-\lambda)^{1-x}dx = C(\lambda)\frac{(\lambda-1)(1-\lambda)^{-x}\lambda^x(x\log(1-\lambda) - x\log\lambda + 1)}{(\log(1-\lambda) - \log\lambda)^2}\Bigg|_{x=0}^{x=1}$$
$$= \frac{\lambda}{2\lambda-1} + \frac{1}{2\tanh^{-1}(1-2\lambda)}$$

$\square$

**Proof of proposition 2 ($\mathcal{CB}$ additional properties)**:

1. Clearly:
$$p(x|\lambda) = \mathbb{1}(x \in [0,1])e^{x\log\frac{\lambda}{1-\lambda} + (\log(1-\lambda) + \log C(\lambda))}$$
So that the continuous Bernoulli indeed forms an exponential family with sufficient statistic $T(x) = x$ and natural parameter $\log(\lambda/(1-\lambda))$. $\square$

2. Also:
$$\text{var}(X) = \int_0^1 C(\lambda)(x - \mu(\lambda))^2\lambda^x(1-\lambda)^{1-x}dx$$
$$= -C(\lambda)\frac{(1-\lambda)^{1-x}\lambda^x\Big(4(\mu(\lambda) - x)\tanh^{-1}(1-2\lambda)\big((\mu(\lambda) - x)\tanh^{-1}(1-2\lambda) - 1\big) + 2\Big)}{\big(2\tanh^{-1}(1-2\lambda)\big)^3}\Bigg|_{x=0}^{x=1}$$
$$= \frac{(\lambda-1)\lambda}{(1-2\lambda)^2} + \frac{1}{(2\tanh^{-1}(1-2\lambda))^2}$$

□

3. Furthermore:
$$F(x) = C(\lambda)\tilde{F}(x|\lambda) = \frac{\lambda^x(1-\lambda)^{1-x} + \lambda - 1}{2\lambda - 1}$$

The above equation, equated to $u$, can easily be inverted algebraically to obtain:
$$F^{-1}(u|\lambda) = \frac{\log(u(2\lambda - 1) + 1 - \lambda) - \log(1 - \lambda)}{\log(\lambda) - \log(1 - \lambda)}$$

□

4. Since $\varphi_X(t) = E[e^{itX}]$, the case where $t = 0$ is trivial. We have:
$$E[\cos(tX)] = \int_0^1 \cos(tx)C(\lambda)\lambda^x(1-\lambda)^{1-x}dx$$
$$= C(\lambda)\frac{(\lambda - 1)(1-\lambda)^{-x}\lambda^x\big((\log(1-\lambda) - \log\lambda)\cos(tx) - t\sin(tx)\big)}{t^2 + \log^2(1-\lambda) + \log^2\lambda - 2\log(1-\lambda)\log\lambda}\bigg|_{x=0}^{x=1}$$

Also:
$$E[\sin(tX)] = \int_0^1 \sin(tx)C(\lambda)\lambda^x(1-\lambda)^{1-x}dx$$
$$= C(\lambda)\frac{(\lambda - 1)(1-\lambda)^{-x}\lambda^x\big((\log(1-\lambda) - \log\lambda)\sin(tx) - t\cos(tx)\big)}{t^2 + \log^2(1-\lambda) + \log^2\lambda - 2\log(1-\lambda)\log\lambda}\bigg|_{x=0}^{x=1}$$

Combining the above two expressions and simplifying, we get:
$$\varphi_X(t) = E[e^{itX}] = E[\cos(tX) + i\sin(tX)] = C(\lambda)\frac{i(1 - \lambda e^{it} - \lambda)}{t + 2i\tanh^{-1}(1 - 2\lambda)}$$

□

5. We have:
$$H(X) = -E[\log p(X|\lambda)] = -E[\log C(\lambda) + X\log\lambda + (1 - X)\log(1 - \lambda)]$$
$$= -\log C(\lambda) - \mu(\lambda)\log\lambda - (1 - \mu(\lambda))\log(1 - \lambda)$$
$$= \mu(\lambda)\log\frac{1-\lambda}{\lambda} - \log C(\lambda) - \log(1 - \lambda)$$

□

6. The KL divergence between $X_1 \sim \mathcal{CB}(\lambda_1)$ and $X_2 \sim \mathcal{CB}(\lambda_2)$ is given by:
$$KL\big(p(x|\lambda_1)||p(x|\lambda_2)\big) = E\left[\log\frac{p(X_1|\lambda_1)}{p(X_1|\lambda_2)}\right]$$
$$= -E[\log C(\lambda_2) + X_1\log\lambda_2 + (1 - X_1)\log(1 - \lambda_2)] - H(X_1)$$
$$= \mu(\lambda_1)\log\frac{\lambda_1(1 - \lambda_2)}{\lambda_2(1 - \lambda_1)} + \log\frac{C(\lambda_1)(1 - \lambda_1)}{C(\lambda_2)(1 - \lambda_2)}$$

□

7. To show that $C(\lambda)$ is convex, we first show that the function:
$$g(\nu) = \begin{cases} \dfrac{\tanh^{-1}(\nu)}{\nu}, & \text{if } \nu \neq 0 \\ 1, & \text{otherwise} \end{cases}$$

is convex in $(-1, 1)$. The Taylor series expansion of $\tanh^{-1}(\nu)$ is given by:
$$\tanh^{-1}(\nu) = \sum_{k=0}^{\infty}\frac{1}{2k + 1}\nu^{2k+1}$$

So that $g(\nu)$ admits the expansion:

$$g(\nu) = \sum_{k=0}^{\infty} \frac{1}{2k+1} \nu^{2k}$$

Then:

$$g''(\nu) = \sum_{k=1}^{\infty} \frac{2k(2k-1)}{2k+1} \nu^{2k-2} \geq 0$$

So that $g$ is convex. Since $C(\lambda) = 2g(1-2\lambda)$, $C(\lambda)$ is convex as well. $\qquad\square$

8. We denote $\tilde{p}(\lambda|\alpha, \beta, \nu) = \lambda^{\alpha-1}(1-\lambda)^{\beta-1}C(\lambda)^{\nu}$. To prove that the C-Beta distribution is a proper distribution, we have to show that the following holds:

$$\int_0^1 \tilde{p}(\lambda|\alpha, \beta, \nu)d\lambda = \int_0^1 \lambda^{\alpha-1}(1-\lambda)^{\beta-1}C(\lambda)^{\nu}d\lambda < \infty$$

While the above integral cannot be computed analytically, we can show that it is indeed finite when $\alpha > 0, \beta > 0$ and $\nu \geq 0$. Let $\epsilon \in (0, 0.5)$ and note that:

$$\int_0^1 \tilde{p}(\lambda|\alpha, \beta, \nu)d\lambda = \int_0^\epsilon \tilde{p}(\lambda|\alpha, \beta, \nu)d\lambda + \int_\epsilon^{1-\epsilon} \tilde{p}(\lambda|\alpha, \beta, \nu)d\lambda + \int_{1-\epsilon}^1 \tilde{p}(\lambda|\alpha, \beta, \nu)d\lambda$$

The middle term above is finite since $\tilde{p}(\lambda|\alpha, \beta, \nu)$ is continuous and is thus bounded in the closed interval $[\epsilon, 1-\epsilon]$. By symmetry of the first and last term, it is enough to prove that the first term is finite (the first and last terms are identical if we switch the roles of $\alpha$ and $\beta$). We use the notation $f_1(\lambda) \lesssim f_2(\lambda)$ to denote that $f_1(\lambda) \leq K f_2(\lambda)$ for some constant $K > 0$ (which does not depend on $\lambda$) for every $\lambda \in (0, \epsilon)$. We have:

$$\tilde{p}(\lambda|\alpha, \beta, \nu) \lesssim \lambda^{\alpha-1}C(\lambda)^{\nu} = \lambda^{\alpha-1}\left(\frac{2\tanh^{-1}(1-2\lambda)}{1-2\lambda}\right)^{\nu} \lesssim \lambda^{\alpha-1}\left(2\tanh^{-1}(1-2\lambda)\right)^{\nu}$$

$$= \lambda^{\alpha-1}\log^{\nu}\left(\frac{2-2\lambda}{2\lambda}\right) = \lambda^{\alpha-1}\left(\log(1-\lambda) - \log(\lambda)\right)^{\nu} \leq \lambda^{\alpha-1}\left(-\log(\lambda)\right)^{\nu}$$

It then follows that:

$$\int_0^\epsilon \tilde{p}(\lambda|\alpha, \beta, \nu)d\lambda \lesssim \int_0^\epsilon \lambda^{\alpha-1}\left(-\log(\lambda)\right)d\lambda \leq \int_0^1 \lambda^{\alpha-1}\left(-\log(\lambda)\right)d\lambda$$

$$= \alpha^{-(\nu+1)}\int_0^\infty e^{-u}u^{\nu}du = \alpha^{-(\nu+1)}\Gamma(\nu+1) < \infty$$

where the second last equality is obtained through the substitution $u = -\alpha\log(\lambda)$. The C-Beta distribution is thus well defined. Finally, the C-Beta clearly satisfies conjugacy, since if $\lambda \sim$ C-Beta$(\alpha, \beta, \nu)$ and $x_1, \ldots, x_n|\lambda \sim \mathcal{CB}(\lambda)$, then:

$$p(\lambda|x_1, \ldots, x_n, \alpha, \beta, \nu) \propto p(x_1, \ldots, x_n|\lambda)p(\lambda|\alpha, \beta, \nu)$$

$$\propto \left(\prod_{i=1}^n C(\lambda)\lambda^{x_i}(1-\lambda)^{1-x_i}\right)\lambda^{\alpha-1}(1-\lambda)^{\beta-1}C(\lambda)^{\nu}$$

$$= \lambda^{\alpha+\sum_i x_i - 1}(1-\lambda)^{\beta+n-\sum_i x_i - 1}C(\lambda)^{n+\nu}$$

$$\propto p(\lambda|\alpha + \sum_{i=1}^n x_i, \beta + n - \sum_{i=1}^n x_i, n + \nu)$$

$\qquad\square$

**Proof of proposition 3 ($\mathcal{CB}$ Bernoulli limit):**

We have that $F(x|\lambda) \xrightarrow{\lambda \to 0} \mathbb{1}(x > 0)$ and $F(x|\lambda) \xrightarrow{\lambda \to 1} \mathbb{1}(x \geq 1)$, which concludes the proof. $\quad\square$

**Proof of proposition 4 ($\mathcal{CB}$ normalizing constant bound):**

It is straightforward to verify, using L'Hôpital's rule, that:

$$\frac{\partial}{\partial \lambda} C(\lambda)\Big|_{\lambda=0.5} = 0$$

Thus, 0.5 is a local optimum of $C(\lambda)$. Since $C(\lambda)$ is convex, $\lambda = 0.5$ is a global minimizer of $C(\lambda)$, and since $C(0.5) = 2$, this finishes the proof. □

**Proof of proposition 5 ($\mathcal{CB}$ maximum likelihood):**

The proof follows from the fact that doing maximum likelihood in an exponential family reduces to doing moment matching on the sufficient statistic. □

## Appendix 4: architectural and training choices

We preprocess MNIST by following the standard procedure of adding uniform $[0, 1]$ noise to the integer pixel values between 0 and 255 and then dividing by 256, resulting in values in $[0, 1]$. For all our MNIST experiments, we use a latent dimension of 20, an encoder with two hidden layers with 500 units each, with $ReLU$ nonlinearities, followed by a dropout layer (with parameter 0.9). The output layer of the encoder has no nonlinearity for the mean and a softplus nonlinearity for the standard deviation. The decoder also has two hidden layers with 500 units, $ReLU$ nonlinearities and dropout, as does the classifier we used to compute the inception score (which has a softmax nonlinearity). The decoder has softplus nonlinearities to enforce nonnegativity (Gaussian standard deviation and beta parameters), sigmoid to enforce values in $(0, 1)$ (continuous Bernoulli, Bernoulli and Gaussian mean). We use a learning rate of 0.001 except for the Gaussian VAE, where we use 0.0001, and optimize with Adam [18] for 100 epochs.

We preprocess CIFAR-10 in the same fashion as MNIST. For CIFAR-10 the latent dimension is 50 and the learning rate for continuous Bernoulli VAE and Bernoulli VAE is 0.001, and 0.0001 for the other distributions. The encoder consists of four convolutional layers, followed by two fully connected ones. The convolutions have respectively, 3, 32, 32 and 32 features, kernel size 2, 2, 3 and 3, strides 1, 2, 1, 1 and are followed by $ReLU$ nonlinearities. The fully connected hidden layer has 128 units and a $ReLU$ non linearity. The decoder has an analogous "reversed" architecture. The classifier used to compute the inception score has a convolution with 32 features and kernel size 3 followed by a $ReLU$ activation, 10 residual blocks at 3 different resolutions and a dense layer with a softmax nonlinearity. Each residual block consists a convolution, $ReLU$, batch normalization, another convolution and adding the result to the input. At the end of each residual block, the resolution is decreased with a convolution with stride two that doubles the number of features, and then dropout (with parameter 0.5) is applied.

## Appendix 5: further experimental results

### 5.1 MNIST

**Inception scores**

In this section we show more inception scores obtained by the continuous Bernoulli VAE and Bernoulli VAE on MNIST, by transforming the decoder with $\mu$ or $\mu^{-1}$ and/or by sampling from $\mathcal{CB}$ or $\mathcal{B}$. The results are in Figure 2

While these inception scores are not what one would normally sample, we can see that the continuous Bernoulli VAE achieves better inception scores than the Bernoulli VAE, regardless of how the decoder is used to obtain the samples. The only situation in which the Bernoulli VAE manages to perform similarly to the continuous Bernoulli is when the decoder is corrected by applying $\mu^{-1}$ after training, similarly to what we observed for the ELBO (although the ELBO is still lower after doing this correction).

**$\beta$-VAE**

As we explained in §4.4 of the main manuscript, the KL term in the ELBO stops the Bernoulli VAE from being interpreted as recovering the mean parametrization of a continuous Bernoulli VAE.

Figure 2: Inception scores for continuous Bernoulli VAE (dark) and Bernoulli VAE (light). Left panel has data (black), $\mathcal{B}(\lambda_{\theta^*(p)})$ (dark orange), $\mathcal{B}(\lambda_{\theta^*(\tilde{p})})$ (light orange), $\mathcal{B}(\mu(\lambda_{\theta^*(p)}))$ (dark purple), $\mathcal{B}(\mu(\lambda_{\theta^*(\tilde{p})}))$ (light purple), $\mu(\lambda_{\theta^*(p)})$ (dark green) and $\mu(\lambda_{\theta^*(\tilde{p})})$ (light green). Right panel has data (black), $\mathcal{CB}(\mu^{-1}(\lambda_{\theta^*(p)}))$ (dark orange), $\mathcal{CB}(\mu^{-1}(\lambda_{\theta^*(\tilde{p})}))$ (light orange), $\mathcal{B}(\mu^{-1}(\lambda_{\theta^*(p)}))$ (dark purple), $\mathcal{B}(\mu^{-1}(\lambda_{\theta^*(\tilde{p})}))$ (light purple), $\mu^{-1}(\lambda_{\theta^*(p)})$ (dark green) and $\mu^{-1}(\lambda_{\theta^*(\tilde{p})})$ (light green).

The $\beta$-VAE [11] recovers disentangled latent representations by optimizing the following objective instead of the ELBO:

$$\mathcal{E}_\beta(p, \theta, \phi) = \sum_{n=1}^{N} E_{q_\phi(z_n|x_n)}[\log p_\theta(x_n|z_n)] - \beta KL(q_\phi(z_n|x_n)||p_0(z_n))$$

where $\beta > 1$ allows to recover disentangled representations. Increasing the weight of the KL penalty makes the mean parametrization interpretation even less viable, as is show in Figure 3:

Figure 3: Effect of $\beta$-VAE when using continuous Bernoulli versus Bernoulli likelihoods.

The figure shows, for $\gamma = 0$, a comparison between $\mathcal{E}_\beta(p, \theta_\beta^*(p), \phi_\beta^*(p))$ and $\mathcal{E}_\beta(p, \theta_\beta^*(\tilde{p}), \phi_\beta^*(p))$ (in dark blue and blue, respectively) and $\mathcal{E}_\beta(p, \theta_\beta^*(\tilde{p}), \phi_\beta^*(p))$ when using $\mu^{-1}(\lambda_{\theta_\beta^*(\tilde{p})})$ as the decoder in light green. We can see that the dark blue and light green curves get further away as $\beta$ increases, meaning that as the representations become more disentangled, the mean parametrization interpretation of the Bernoulli VAE becomes even worse. While not directly noticeable on the figure, the gap between the dark and light blue curves also increases, from 304 to 339, meaning that the performance of a Bernoulli VAE decreases as $\beta$ is increased.

**Flexible approximating posterior**

Here we reproduce our experiments on the warped MNIST datasets from the main manuscript, but we use a more flexible approximate posterior using 10 layers of planar flows [33]. Results are in Figure 4, where we can see that not only does the continuous Bernoulli still outperform the Bernoulli, but the gap becomes even larger.

Figure 4: Continuous Bernoulli VAE ELBO comparison against Bernoulli VAE with normalizing flow approximate posterior.

Figure 5: MNIST continuous Bernoulli VAE and Bernoulli VAE samples 1. Highlighted in red are shown in the main manuscript.

**Samples**

In this section we show samples used to compute the inception scores for MNIST (for warping $\gamma = 0$, that is, without transforming the data). Samples can be seen in Figures 5 through 8.

**5.2 CIFAR-10 samples**

In this section we show samples used to compute the inception scores for CIFAR-10 in Figures 9 through 11.

$\mathcal{CB}(\mu(\lambda_{\theta^*(p)}))$  $\mathcal{CB}(\mu(\lambda_{\theta^*(\tilde{p})}))$  $\mathcal{B}(\lambda_{\theta^*(p)})$  $\mathcal{B}(\lambda_{\theta^*(\tilde{p})})$  $\mathcal{B}(\mu(\lambda_{\theta^*(p)}))$  $\mathcal{B}(\mu(\lambda_{\theta^*(\tilde{p})}))$

Figure 6: MNIST continuous Bernoulli VAE and Bernoulli VAE samples 2.

$\mu^{-1}(\lambda_{\theta^*(p)})$  $\mu^{-1}(\lambda_{\theta^*(\tilde{p})})$  $\mathcal{CB}(\mu^{-1}(\lambda_{\theta^*(p)}))$  $\mathcal{CB}(\mu^{-1}(\lambda_{\theta^*(\tilde{p})}))$  $\mathcal{B}(\mu^{-1}(\lambda_{\theta^*(p)}))$  $\mathcal{B}(\mu^{-1}(\lambda_{\theta^*(\tilde{p})}))$

Figure 7: MNIST continuous Bernoulli VAE and Bernoulli VAE samples 3.

$\mathcal{N}$ / $p$  $\mathcal{N}$ / $\tilde{p}$  $\mathcal{N}$ **mean** / $p$  $\mathcal{N}$ **mean** / $\tilde{p}$  $B$ / $p$  $B$ / $\tilde{p}$  $B$ **mean** / $p$  $B$ **mean** / $\tilde{p}$

Figure 8: MNIST Gaussian VAE (denoted $\mathcal{N}$) and beta distribution VAE (denoted $B$) samples, both including normalizing constants (denoted $p$) and ignoring them (denoted $\tilde{p}$). Highlighted in red is shown in the main manuscript.

**data** $\quad f_{-0.5}(\textbf{data}) \quad \lambda_{\theta^*(p)} \quad\quad \lambda_{\theta^*(\tilde{p})} \quad \mathcal{CB}(\lambda_{\theta^*(p)}) \quad \mathcal{CB}(\lambda_{\theta^*(\tilde{p})})$

Figure 9: CIFAR-10 continuous Bernoulli VAE and Bernoulli VAE samples 1. Highlighted in red corresponds to the highlighted MNIST samples.

$\mu^{-1}(\lambda_{\theta^*(p)}) \quad \mu^{-1}(\lambda_{\theta^*(\tilde{p})}) \quad \mathcal{CB}(\mu^{-1}(\lambda_{\theta^*(p)})) \quad \mathcal{CB}(\mu^{-1}(\lambda_{\theta^*(\tilde{p})}))$

Figure 10: CIFAR-10 continuous Bernoulli VAE and Bernoulli VAE samples 2.

Figure 11: CIFAR-10 Gaussian VAE (denoted $\mathcal{N}$) and beta distribution VAE (denoted $B$) samples, both including normalizing constants (denoted $p$) and ignoring them (denoted $\tilde{p}$). Highlighted in red corresponds to the highlighted MNIST samples.