[Reviews · NeurIPS 2019]

Reviewer 1



I have read the author response and am strengthening my confidence (since after the response and seeing other reviews, I believe I have understood everything correctly). I loved the analysis of the "concentration of mass at the extrema" between the CB and beta that the authors provided in their response. It is exactly this kind of careful study and how it relates to what you saw in your experiments with MNIST (and why it matters specifically for the particular characteristics of the dataset) that make me love a paper like this. I hope that the authors add that analysis to the supplementary material at least. It almost sounds like your supplement could even be a mini paper on such a study that's interesting in its own right (though please don't write another paper on it). *** The paper is very classy and very well written. I find the paper very inspiring. I will fight for the paper to be accepted if needed. I do believe in the significance/impact of the paper as a demonstration of how to carefully study your modeling choices, and possible negative repercussions of some choices, from a technical perspective. I loved things like studying the shape of the normalising constant (as a function of the parameter). I would like researchers to more often mimic such studies in their own work. However, I don't think the paper is necessarily "seminal" as I can't see many people opting to use this new continuous Bernoulli distribution instead of rather nicer distributions on the unit interval, namely, the beta distribution. With that said, why did Kingma and Welling not just use the beta distribution as the likelihood in the first place (in the same way you defined in your experiments section)? It looks like empirical performance wise they have comparable/mixed performance, and qualitatively (the samples of the digits) of the CB is more satisfying. Is that correct? I see that this is reviewed but I wish it were emphasised more in other parts of the paper, like in the introduction. I don't think it will weaken the paper. I think the CB distribution could be interesting in its own right. But again it's rather ugly compared to the beta distribution. I think the way to go would be a study of the qualities/differences between the beta and CB, for example, comparing the shapes of their normalising constants. Or the metrics of wanting to directly model a the single parameter of the CB (and its interpretations). But I understand this is beyond the scope/needs of the paper. Can't say I found anything lacking in the paper.

Reviewer 2



updates after rebuttal ---------------------------- I read the rebuttal and still think more evidences are needed to prove that continuous bernoulli is a right choice. I agree with the other reviewers that this is an interesting and valid question to look into. However, I do find in the current form this paper does not meet the requirements to be published at NeurIPS. - The simplicity of the approach is not necessarily a problem, but in this case we would like to see more empirical analysis (ideally on real-world architectures used every day and common benchmarks). This is clearly missing from the current paper. - There is no clear evidence that the proposed continuous Bernoulli distribution outperforms other easy or widely used choices (e.g, Beta, Gaussian, 256-way softmax, discretized logistic). In the newly added experiements on CIFAR10 the proposed CB distribution is outperformed by Beta, which obviously leads to the concern about the motivation. Several comments: 1. There isn’t much information in section 4.1, 4.3. 2. In section 4.5, it is mentioned that applying the transformation of the mean of continuous Bernoulli can help fix some problem of the VAE trained with improper bernoulli likelihoods. It is pointed out by the authors that the common reasoning is false but interestly, there is still improvements if we do this. Since this is more an analysis paper, it would be more interesting to see the reason. 3. The results on MNIST is good and clearly show improvements. But given the idea is rather simple, I’m expecting more experimental evidence on SOTA architectures. The existing results on CIFAR is not satisfying. 4. The authors show in some experiments that continuous Bernoulli has better performance than Beta distribution, which is a natural choice for continuous distribution over [0, 1]. But there is no explanation for this. Why should we prefer continuous Bernoulli over Beta?

Reviewer 3



## The continuous Bernoulli: fixing a pervasive error in variational autoencoders. ### Summary The authors propose a new single-parameter distribution with support on the open interval (0,1). This distribution is the result of properly normalizing a Bernoulli distribution when we allow the random variable to take values in the continuous interval (0,1) rather than just in the discrete set {0,1}. This so called "Continuous Bernoulli" distribution belongs to the exponential family and have several nice properties like reducing to the classic Bernoulli distribution in the limit. In the experiments, the authors apply it as the likelihood of a VAE to model image datasets where the intensity of the pixels are normalized between 0 and 1. Previous work in the literature have model this incorrectly by using a Bernoulli likelihood, and so the authors show that using a proper distribution (the propose continuous version of the Bernoulli distribution) improves the performance. ### Details The motivation of the paper is simple and the exposition is clear. In addition, it is really well-written which raises the quality of the paper. Some sentence however should be slightly tone down (just a minor). The main idea of the paper is to propose a new distribution which is simply a continuous version of the Bernoulli distribution. This is done by assuming the same functional form of the Bernoulli distribution and then computing the right normalization constant in closed from. Surprisingly, and to the extend of my knowledge, this distribution has not been previously proposed in the literature. The authors provide a through analysis of this new distribution computing the pdf as well as the first moment, showing that it belongs to the exponential family, that it is amenable to the re-parameterization trick and that in the limit it converges to the classical Bernoulli distribution (among other properties). One derivation that it is missing and I thing it is pretty important in better understanding the properties of a distribution is the moment generating function that has closed form (see at the bottom the derivation). In the experiments the authors, the authors apply it as the likelihood in the VAE and use it to model an image dataset where the support is bounded. This was previously done by incorrectly using the Bernoulli distribution ignoring the fact that the observations were not binary but rather continuous values between 0 and 1 after renormalizing the pixel intensities to fall in this interval. They show that the propose continuous Bernoulli distribution outperforms the VAE with Bernoulli, Gaussian and Beta likelihoods. Maybe, the weakest point of the paper is that the fail to further analyses why the Beta shows a worse performance. The Beta distribution is a more flexible object that the proposed distribution in that it has two parameters and that can model multi-modal distributions. Given that, I would expect to see a trade off between the two distributions (not one outperforming the other in all situations). The Beta distribution have the advantage of being more flexible, but at the same time it would be more prone to over fitting due to a higher number of parameter. Additionally, the beta distribution is tricky to optimize due to the gamma function in the normalization constants, and maybe the proposed distribution has a better behavior. This analysis is missing in the paper. Despite this, the paper is well-motivated, they propose a simple approach to a prevalent problem in the previous literature and has a significant novelty by introducing a distribution that despite its simplicity, and to the extend of my knowledge, has not been proposed before. ## Moment generating function Just applying integration by parts you can get that the MGF = \frac{2 \tanh^{-1}(1-2\lambda)}{2 \tanh^{-1}(1-2\lambda) - t} \frac{\lambda \exp(t) + \lambda -1}{2\lambda-1} (for \lambda \neq 0.5).

[Author Response · NeurIPS 2019]

**Paper 7283 | The continuous Bernoulli: fixing a pervasive error in variational autoencoders**

We sincerely thank the reviewers for helping us improve this work, as well as for finding it "inspiring" and "very well
written". We will make all minor changes requested, and here we focus on the major points of feedback.

• **More attention/detail to continuous Bernoulli vs beta (raised by all reviewers)**: Thank you, this is important.
There are two main points; we will add a thorough discussion of them, and new results, to the paper.

1. As a device for study: the continuous Bernoulli is the natural and necessary distribution to understand the impact
of erroneously using a Bernoulli likelihood on $[0, 1]$-valued data, thus providing critical insight into one main and
important question raised in this work.

2. Concentration of mass at the extrema: the notably different behavior of beta and continuous Bernoulli near
the extrema warrant observation. Denoting $p(x|\lambda)$ and $p(x|\alpha, \beta)$ as continuous Bernoulli and beta densities,
respectively, we can show that $\lim_{x \to 0} p(x|\lambda)/p(x|\alpha, \beta) \to 0$ if $\alpha < 1$, and $\lim_{x \to 0} p(x|\lambda)/p(x|\alpha, \beta) \to \infty$ if
$\alpha > 1$; a finite positive limit is only achieved if $\alpha = 1$. (An analogous result holds as $x \to 1$ depending on
the sign of $\beta$, but we focus on a neighborhood of 0 here.) Next, for a given $p(x|\lambda)$, denote $p(x|\alpha_\lambda, \beta_\lambda)$ as the
corresponding beta distribution with matching mean and variance. Then, $\lambda < 0.5$ implies $\alpha_\lambda < 1$, which in other
words shows that, for comparable moments, the beta distribution places much more mass in a neighborhood of
0 (see Figure 1 below). Note that this does not imply that the continuous Bernoulli is not placing most of its
mass around 0, just not as much as the beta. This key insight highlights that even "similar looking" continuous
Bernoullis and betas behave considerably differently at the extrema, which is precisely the most important part of
the densities when modeling almost binary data (such as MNIST, which while almost binary, does have grayscale
pixels). Empirically, we find that the beta distributed VAE produces means that are *less* extremal (i.e. grayer
images, as seen in the appendix materials), which implies that each beta was shifted away from 0 and 1 to reduce
adding too much mass to the extrema (precisely the effect of Figure 1).

• **State of the art architectures (reviewer 2)**: To further show the meaningful performance gains of the continuous
Bernoulli, we have run additional experiments with a normalizing flow architecture in the approximate posterior
(Rezende and Mohammed, 2015). Figure 2 (below) shows that not only does the continuous Bernoulli still outperform
the Bernoulli, but the gap is even larger than in simpler architectures. For CIFAR-10 the beta distribution outperforms
the continuous Bernoulli, which we suspect to be because beta can concentrate mass anywhere in RGB space.
However, even in this case, the continuous Bernoulli outperforms the Bernoulli in a non-marginal way: again,
ignoring normalizing constants significantly hurts performance. This gives an exciting first answer to this comment,
and we will include more experiments with more complicated architectures on CIFAR-10 by publication time. Of
course, the advantages of the continuous Bernoulli for data with values close to the extrema might suggest future
distributions more suitable for CIFAR-10, which we will also explore.

• **More analysis of the continuous Bernoulli distribution (reviewer 3)**: As you pointed out, the continuous Bernoulli
distribution has a closed form moment generating function; thank you. We have computed its characteristic function,
entropy, and the KL divergence between two continuous Bernoullis, all of which also have closed form expressions
(omitted here due to space constraints, but we will include them in the paper). These additions will make for a very
thorough characterization of our distribution.

• **Why $\mu^{-1}$ does not equate the Bernoulli to the continuous Bernoulli (reviewer 2)**: In a maximum likelihood
setting with no latent variables, $\mu^{-1}$ does offer an equivalence. While one might expect this to fully carry over in a
setting with latent variables, that is incorrect; the full argument for which is presented in section 4.5 of the paper.
Even still, using $\mu^{-1}$ still recovers some amount of the lost performance. Is this finding a fundamental suboptimality
of the Bernoulli model, or simply an artifact of approximate inference (training with the ELBO)? We have run the
following additional experiment: we use the EM algorithm to estimate the parameters of a mixture of continuous
Bernoullis (loosely, the VAE can be thought of as an infinite-component extension) on simulated data. Figure 3
(below) shows that correcting with $\mu^{-1}$ still does not fully correct for using a Bernoulli likelihood (as measured by
distance between the estimated distribution and the ground truth). This is thus a fundamental modeling issue.

Figure 1                              Figure 2                              Figure 3

[Meta-Review · NeurIPS 2019]

This paper generated an incredible amount of discussion among the reviewers, with many "pros": -- The paper identifies a bad practice that so many others have not so carefully dealt with in the past. -- The paper addresses it not by simply throwing 20 different modeling choices and comparing them, but instead choosing one, the Continuous Bernoulli, and analyzing what happens when you apply it to MNIST. The paper asks the question: "if we assume as others before that we may treat as binary, are the bad implications negligible?" The paper shows that the answer is very much no by exploring the shape of the normalising constants and displaying a logical, scientifically exposited train of thought to precisely characterise the source of the resulting error. -- The experimental section is enough to show the benefits of this likelihood. Adding experiments with new architectures would not give meaningful insights since it is a kind of independent choice. The reviewers would ask the authors to carefully incorporate this question and variants of this question in their final version: "If a Gaussian likelihood has a support mismatch, then just truncate the Gaussian on (0, 1), why not this choice?" and clearly acknowledge that we already have a bunch of other choices (which have been justified in practice). From an area chair's point of view, I would like to point out that the paper has a tone of "slacking" other VAE papers, as if the other authors were sloppy and made errors unwittingly. One has to be cautious here. I read the papers that are referred to, and don't actually believe that any of these papers contain a "pervasive error" and would come to their defense. Instead, many of these papers use Binary MNIST as a benchmark simply because of the readily available list of other Binary MNIST benchmarks -- people already know in their sleep how many nats constitute a good Binary MNIST benchmark. Therefore: you introduce a nice new distribution, which stands as a piece of work in its own right. Why couple its "sales pitch" to disparaging the works of others without a good enough justification? Why not just introduce the distribution, saying that your C() log normalizing term would fix the Bernoulli if people weren't careful, and clearly point out to the community why one has to be careful if the TensorFlow Bernoulli classes aren't typed as Boolean but as floats? For instance, a title like "The Continuous Bernoulli Distribution" or "The Continuous Bernoulli Distribution for _____" would also be okay as your work applies to any sloppily used Bernoulli likelihood, not only in its use in VAEs? And the point that you really make is that there is (unfortunately) room to be sloppy as the Bernoulli loss, as implemented in many libraries, doesn't take a single "bit" or "boolean" as data type...